# HyPedSim: A Multi-Level Crowd-Simulation Framework—Methodology, Calibration, and Validation [note 1]

**DOI:** 10.3390/s24051639

**Published:** 2024-03-02

**Authors:** Huu-Tu Dang, Benoit Gaudou, Nicolas Verstaevel

**Affiliations:** UMR 5505 IRIT, Université Toulouse Capitole, 31000 Toulouse, France; benoit.gaudou@ut-capitole.fr (B.G.); nicolas.verstaevel@ut-capitole.fr (N.V.)

**Keywords:** agent-based model, multi-level behaviour, pedestrian modelling, multi-scale simulation

## Abstract

Large-scale crowd phenomena are complex to model because the behaviour of pedestrians needs to be described at both strategic, tactical, and operational levels and is impacted by the density of the crowd. Microscopic models manage to mimic the dynamics at low densities, whereas mesoscopic models achieve better performances in dense situations. This paper proposes and evaluates a novel agent-based model to enable agents to dynamically change their operational model based on local density. The ability to combine microscopic and mesoscopic models for multi-scale simulation is studied through a use case of pedestrians at the Festival of Lights, Lyon, France. Pedestrian outflow data are extracted from video recordings of exiting crowds at the festival. The hybrid model is calibrated and validated using a genetic algorithm that optimises the match between simulated and observed outflow data. Additionally, a local sensitivity analysis is then conducted to identify the most sensitive parameters in the model. Finally, the performance of the hybrid model is compared to different models in terms of density map and computation time. The results demonstrate that the hybrid model has the capacity to effectively simulate pedestrians across varied density scenarios while optimising computational performance compared to other models.

## 1. Introduction

Pedestrian simulation is an effective tool to model and study the behaviour of pedestrians in various contexts, from optimisation of pedestrian flows to safety issues [1]. Pedestrian behaviour can commonly be modelled into a three-layer level architecture [2]:Strategic level: pedestrians determine a list of activities (or targets) and when they want to perform these activities.Tactical level: pedestrians choose a path to the predefined destinations based on information about the environment.Operational level: pedestrians adjust their local movements such as collision avoidance to adapt to the surrounding area.

In order to build a pedestrian simulation, modellers have to effectively instantiate their model for at least one of the three layers, that is to say at least one strategic model (also referred to as goal selection model) [3,4], or one tactical level model (also known as path planning model) [5,6], or one operational level model [7,8]. The majority of research contributes to the operational level because this level addresses the immediate, physical behavior of pedestrians such as collision avoidance or local movements. This paper focuses on selecting the proper algorithm for the operational level which computes the velocities and speed of each pedestrian based on local information.

Many types of operational models exist in the literature, each model being built to capture and mimic a specific phenomenon [9]. Therefore, their performance differs if they are put into contexts for which they have not been designed. Microscopic models are usually used to simulate pedestrians in low- and medium-density situations, where the density level is less than 2 ped/m^2^ [1]. These models encounter specific problems when applied to high-density situations, such as abnormal vibrations with the Social Force Model [10], congestion in dense bidirectional flow with the Velocity Obstacle Model [11], and unrealistic results in terms of collision metrics with the data-driven models [12,13]. On the other hand, pedestrian dynamics in high-density situations are similar to fluid flow [14], and fluid-like equation models [14,15] are particularly appropriate for these scenarios due to their assumption of the continuousness of pedestrian flow. Given the observed variability in the performance of pedestrian-simulation models across different crowd densities, the integration of multiple models is required to accommodate the full spectrum of potential scenarios.

This contribution proposes and evaluates an agent-based model to enable agents to dynamically change their operational model based on local density to adapt to their dynamic environments. An agent-based approach is selected as it is a common method to model pedestrians in which agents represent pedestrians with decision-making capabilities, personal characteristics, and varied levels of operations and interactions [16]. Local density estimation is achieved through predefined regions in the environment, which enables contextual and structured adaptation to various environments. Additionally, the use of predefined areas can facilitate the implementation of the model by providing a more intuitive and manageable approach to estimating density. The ability to combine microscopic and mesoscopic models for multi-scale simulation is studied through a use case of pedestrian crowds at the Festival of Lights, Lyon, France—the so-called HyPedSim framework. This case study demonstrates the utility of the framework with respect to integrating different models in order to enable festival organisers to test various experiments of large-scale crowd simulation for safety planning and control.

The paper is organised as follows. Related works are summarised in Section 2. Section 3 describes our proposed agent-based model for the multi-level behaviour of pedestrians. Next, Section 4 illustrates the ability to couple models through the applications of simulating pedestrian crowds at the Festival of Lights. The performance of the hybrid model in comparison to other models is then demonstrated in Section 5. Finally, the paper ends with a conclusion and discussion in Section 6.

## 2. Related Work

Hybrid modelling is a methodology that incorporates different modelling techniques into a single framework. In pedestrian modelling, a hybrid model usually integrates macroscopic or mesoscopic models with microscopic models [17]. This combination enables the hybrid model to exploit the strengths of both approaches, thereby achieving a balance between simulating global crowd dynamics with reasonable computational cost while preserving the detailed representation of individual behaviour.

The classification of hybrid models relies on the communication mechanism between different simulation models, which can be categorised into two main types: auto-switch models and region-based models. Auto-switch models [18,19,20] allow only one model to run at a time but can switch from a macroscopic model to a microscopic model and vice versa based on a trigger condition. In contrast, the region-based model [21,22] assigns certain areas in the environment for each model and runs these models concurrently. A mechanism for the aggregation and disaggregation of crowds is required for hybrid models to have a consistent transition between the macroscopic and microscopic levels.

On the other hand, the calibration of pedestrian-simulation models is an essential step to improve the realism of simulation results. This step involves fine-tuning the parameters of a pedestrian-simulation model to ensure that simulation outputs match observed data [23]. Various techniques have been used to calibrate simulation models such as genetic algorithm [24], Bayesian inference [25], and maximum likelihood estimation [26]. However, calibrating hybrid models requires greater complexity, as there are more parameters across multiple constituent models. Another difficulty specific to crowd models is the lack of empirical pedestrian data in high-density situations due to challenges in accurately extracting pedestrian features from video recordings of crowds [27].

Currently, there is a lack of a general framework for modelling multi-level pedestrian behaviour that allows agents to dynamically change their operational models. Although Curtis et al. [28] proposed the Menge framework, a crowd simulator that aggregates different modelling algorithms for each level of pedestrian behaviour, this framework only allows one modelling technique to be implemented at the operational level in the simulation. To tackle this challenge, our work focuses on developing an agent-based model that simulates multi-level pedestrian behaviour and is compatible with various modelling algorithms at each level. Furthermore, this model enables pedestrian agents to switch their operational models in response to changes in local density. A detailed description is provided in the following section.

## 3. HyPedSim Framework

This section presents HyPedSim, a framework that allows agents to dynamically change their operational models. Note that while our primary focus is at the operational level, the tactical level component is also described to demonstrate that this approach is extensible to model the higher levels of pedestrian behavior.

### 3.1. General Overview

The environment is divided into different zones, with each zone comprising walkable space and potential obstacles. This division of the environment can be based on expert knowledge or particular crowd characteristics and environmental criteria. Figure 1 illustrates an example of the environment separated into four distinct zones while a concrete example of zones is displayed in Section 4. An appropriate pedestrian-simulation model, such as the Social Force Model [7] or Continuum Crowds Model [15], is selected to simulate pedestrians in each zone depending on crowd density. Each zone is associated with specific triggering rules and transition functions.

The triggering rules evaluate pedestrian and environmental characteristics within the respective zone to select suitable models for pedestrian simulation. Due to the varying performance of pedestrian models across different crowd densities, these rules identify appropriate models to accurately capture the dynamics within each area. When new individuals enter a zone, the transition functions are activated to adapt their information, facilitating a smooth transition from their previous zone to the current one.

When entering a new zone, pedestrians adjust their tactical and operational models to align with the requirements of the corresponding zone. To ensure a generic and extensible approach for various pedestrian-simulation models, an agent-based model is proposed in Section 3.2 to model the multi-level behaviour of pedestrians. This agent-based model enables pedestrians to dynamically select the appropriate modelling algorithm at each level of behaviour.

### 3.2. Agent-Based Model for Multi-Level Behaviour

To develop an agent-based model for multi-level pedestrian behaviour, an abstraction is first introduced to describe the problem of pedestrian modelling. The problem is divided into two subproblems: *tactical* and *operational subproblem* (the integration of a *strategic subproblem* is not in the scope of this paper). Suppose that S={E,A}, where *E* and *A* are the environment and the set of agents in simulation, respectively. The *tactical subproblem* can be formalised by the function *T,* $T:S\times t\times R^{2}\to R^{2}$, which maps the simulation state, time, and the agents’ destination to the local target which is in $R^{2}$. The *operational subproblem* can be formalised as follows: $O:S\times t\times R^{2}\times \{0,1\}\to R^{2}$. The function *O* maps the local simulation state *S*, time, and local target, and a binary value (1 if agents are in high-density zones or 0 otherwise) to a feasible velocity in $R^{2}$, which is then used to update agents’ locations in the next simulation step. In general, the problem of pedestrian modelling can be stated via the following mathematical formulation:(1)$$\begin{array}{cc} v_{i}\left(t\right) & =O_{i}\left(T_{i}\left(t\right)\right) \end{array}$$(2)$$\begin{array}{cc} p_{i}\left(t+1\right) & =p_{i}\left(t\right)+v_{i}\left(t\right)\Delta t \end{array}$$
where $v_{i}\left(t\right)$ and $p_{i}\left(t\right)$ are the velocity and the location of agent *i* in the xy-coordinate system at time *t*, respectively, and Δt is the duration of a simulation step.

Given the description above, an agent-based model is designed as shown in Figure 2. To include different modelling algorithms, each subproblem is represented by an abstract class (“class” refers to the classical definition in the Object Oriented Programming paradigm). The *tactical subproblem* is presented via the *Tactical level* while the *Operational level* class presents the *operational subproblem*. Each abstract class can contain different inheritance subclasses, where each subclass represents a specific model to simulate various behaviours depending on the situation (strategy pattern). The *Pedestrian* class has two attributes, which are tactical_level and operational_level, that are responsible for communicating with the *Tactical level* and *Operational level*. The pedestrian agents are only allowed to present one model for each level of behaviour at a time, but that model may be changed based on the state of the environment in which the pedestrian agents are operating.

Each modelling algorithm for each level needs to inherit the corresponding abstract class. The *Tactical level* and *Operational level* classes return the local target and feasible velocity for pedestrians, respectively. What should be noted here is that in the simulation environment, excluding the *Pedestrian* class, only one single instance of each class is created (singleton pattern). Therefore, our model does not increase complexity but improves the flexibility of simulation. Furthermore, it is possible to change the modelling algorithm at each level of behaviour by simply updating the variables tactical_level and operational_level in the *Pedestrian* class.

### 3.3. Pedestrian Activity Diagram

Figure 3 illustrates the activity diagram of pedestrian agents at each simulation step. Initially, the agents perceive their surrounding environment to identify neighbouring agents. Subsequently, if the agents have either reached their current local target or do not have a local target, they execute *define local target* behaviour which invokes their corresponding inheritance model from the *Tactical level* class to assign a new local target. Afterward, pedestrian agents perform the *move* behaviour using the returned velocity from their respective inheritance model in the *Operational level* class.

After moving to new locations, pedestrians check whether they have arrived at their destination. If yes, their movement is completed. Otherwise, they determine if they have transitioned into a new zone. In such cases, pedestrian agents need to update their states by sending a request to the zone entity to obtain information about the tactical and operational models employed in this new zone. This query returns the results $tactical\_level_{new}$ and $operational\_level_{new}$, which represent the operational and tactical in use within the zone, respectively. Pedestrians then update their attributes by setting their operational variable operational_level to $operational\_level_{new}$ and their tactical variable tactical_level to $tactical\_level_{new}$.

This continuous process of perceiving the environment, defining local targets, moving, and updating states based on zone transitions enables pedestrian agents to navigate through the simulation space effectively. This dynamic approach allows for the modelling of complex scenarios and interactions between agents, accounting for varying tactical and operational models across different zones.

## 4. Application to the Festival of Lights

This section demonstrates the capacity of the HyPedSim framework to model large, dense crowds in mass-gathering events such as the Festival of Lights. The festival is first introduced, followed by a discussion of empirical data collection. The pedestrian-simulation models and their coupling used in the HyPedSim framework are then described in detail. Finally, the calibration of the framework using a genetic algorithm along with the validation and sensitivity analysis of the calibration results are presented.

### 4.1. Festival of Lights

The Festival of Lights [29] is an annual art festival in Lyon, France. During the Festival of Lights, visitors gather to watch light shows projected onto the walls of the city’s most beautiful monuments. Figure 4 shows pedestrians watching a light show in Place des Terreaux (as highlighted by the red rectangle), the festival’s central plaza and most crowded area. After the show ends, pedestrians head towards the two pre-configured exit roads (Constantine Road and Chenavard Road) designed for safety management, as illustrated via the bottom-left blue arrows.

### 4.2. Data Collection

Two cameras are used to record the crowd exiting, with each camera corresponding to one exit road. Subsequently, the pedestrian outflow on each road is manually calculated. Figure 5 shows the instant outflow of pedestrians over time on Constantine and Chenavard Roads. A total of 3833 pedestrians are counted, with 1803 pedestrians on Constantine Road and 2030 pedestrians on Chenervard road. The exit time of the crowd is almost 400 s. The Gaussian filter method with a standard deviation of 2.0 for the Gaussian kernel is applied to reduce fluctuation error and improve the consistency of the outflow.

### 4.3. Pedestrian-Simulation Models

During the pedestrian exit process, the plaza experiences high density, while the two exit roads exhibit lower density. Our proposed agent-based model can effectively simulate these situations where density varies spatially and temporally. To achieve this, two pedestrian-simulation models—the Social Force Model [7] and the Continuum Crowds Model [15], as well as their hybridisation in HyPedSim—are incorporated at the operational level of the framework to simulate dense crowd exit scenarios at the Festival of Lights.

#### 4.3.1. Social Force Model

This section presents the simplified Social Force Model (SFM) [7] used in the framework, which describes pedestrian motion as driven by social forces. These forces result from both internal factors, such as an individual’s attraction towards a personal goal, and external factors, like repulsion from neighbouring pedestrians. The SFM is formulated based on Newton’s second law:(3)$$a_{i}\left(t\right)=\frac{v_{i}^{pref}-v_{i}\left(t\right)}{\tau }+\sum_{j\in N(i)}n_{ij}\left(t\right)A\exp^{-\frac{d_{ij}\left(t\right)}{B}}$$
where $a_{i}\left(t\right),v_{i}\left(t\right)$ are the acceleration vector and velocity vector of pedestrian *i* at time *t*, respectively. Let $v_{i}^{pref}$ denote preferred velocity of pedestrian *i*, and $V^{pref}=∥v^{pref}∥$ represent the preferred speed of pedestrians. The first term of Equation (Equation 3) accounts for the acceleration from the current velocity to the preferred velocity within the reaction time τ. The second term in Equation (Equation 3) describes distance-based repulsion forces with other pedestrians in the neighbour set N(i), where A,B are the parameters controlling the strength and range of the repulsion forces and:(4)$$d_{ij}\left(t\right)=∥x_{i}\left(t\right)-x_{j}\left(t\right)∥-\left(r_{i}+r_{j}\right)\;\;\mathrm{and}\;\;n_{ij}\left(t\right)=\frac{x_{j}\left(t\right)-x_{i}\left(t\right)}{∥x_{i}\left(t\right)-x_{j}\left(t\right)∥}$$
where $x_{i}\left(t\right)$ and $r_{i}$ are the position at time *t* and radius of pedestrian *i*, respectively. $d_{ij}\left(t\right)$ is the separation distance between pedestrian *i* and *j*, and $n_{ij}\left(t\right)$ is the unit vector pointing from $x_{j}\left(t\right)$ to $x_{i}\left(t\right)$. Figure 6 illustrates intuitively the repulsion force $f_{ij}$ between pedestrian *i* and *j*.

The SFM has been demonstrated to realistically reproduce self-organisation phenomena in low-density situations such as lane formation and arc-shaped clogging at bottlenecks [7]. There are four parameters of the SFM for calibration including: $A,B,V^{pref},\tau$.

#### 4.3.2. Continuum Crowds Model

On the other hand, Treuille et al. [15] proposed the Continuum Crowds (CC) Model, which treats pedestrian flow similarly as a continuum without taking into account individual differences. Global navigation is managed via dynamic potential field using the eikonal equation (Equation (Equation 5)): (5)$$\begin{array}{ccc} ∥\nabla \phi (x)∥ & =C & \end{array}$$(6)$$\begin{array}{cc} \;\;\;\;\;\;\;\;\;\;\;\;\;\;\;\;\;\;\;\;\;\;\;\;\;\;\;\;\;\;\;\;\;\;\;\;v & =-f\left(x,\theta \right)\frac{\nabla \phi (x)}{∥\nabla \phi (x)∥} \end{array}$$
where ϕ represents the potential function and *C* denotes the estimated unit cost in the direction of the gradient ∇ϕ. The cost value to the goals is evaluated based on distance, time, and discomfort factors. Pedestrian velocity is determined by the direction opposite the gradient, with the magnitude depending on the speed field f(x,θ) evaluated at position x with moving direction θ.

The CC model is suitable for simulating large crowds in extreme-density situations with computational efficiency. To apply the CC model, the environment is discretised into cells. Pedestrians in the same cells, with the same target, have the same velocity. The magnitude of velocity in each cell, assuming that the environmental topography is characterised with no effect of slope, is defined based on local density:(7)$$\begin{array}{c} f\left(x,\theta \right)=f_{max}-\frac{\rho \left(x+rn_{\theta }\right)-\rho_{min}}{\rho_{max}-\rho_{min}}\left(f_{max}-f_{flow}\left(x,\theta \right)\right) \end{array}$$
where $\rho_{max},\;\rho_{min}$ are the maximum and minimum density thresholds affecting the speed value, respectively. $f_{flow}\left(x,\theta \right)$ represents the average flow speed in the direction θ at the position x. Likewise, $\rho (x+rn_{\theta })$ denotes the local density evaluated at the point $x+rn_{\theta }$ which is at distance *r* from x in the direction θ.

Figure 7 describes the relationship between density and pedestrian speed. Pedestrian speed reaches its maximum value when the density is below $\rho_{min}$. In contrast, if the density exceeds $\rho_{max}$, pedestrian speed is limited to the average flow speed of crowds. The minimum value of average flow speed is set at $f_{min}$. Between these density thresholds, pedestrian speed linearly decreases from its maximum value to its minimum value. Therefore, the parameters that need to be calibrated are: $f_{min},f_{max},\rho_{min},\rho_{max}$.

#### 4.3.3. Hybrid Model

Pedestrian agents follow the activity diagram depicted in Figure 3 in general, with additional details when specifically applied to simulating crowds at the Festival of Lights.

**Specification of operational level models**: During the pedestrian exit process at the Place des Terreaux, the plaza experiences high density, while the two exit roads exhibit lower density. To handle these multi-density dynamics, the environment is divided into three distinct zones (as illustrated via the three rectangles in Figure 4). The plaza is specified as a high-density zone (red rectangle) and two exit roads are specified as low-density zones (blue rectangles). Each zone is characterised by one specific operational level model for pedestrian simulation. The specifications of the operational level models for these zones are as follows:The CC model [15] for a single target cell is used to simulate pedestrians in the high-density zone due to its effectiveness in dense scenarios. This approach leads to further discretisation into cells, each storing information about the environment and the pedestrians, such as average velocity and local density.The SFM [7] is applied to the two low-density zones to simulate pedestrians who have exited the plaza to one of the two exit roads, as it can realistically simulate pedestrians in low-density situations.

**Criteria to transition operational level models**: The transition criteria are activated as soon as pedestrian agents move out of the high-density zone into a new, low-density zone. When this occurs, the transition rule is applied, which sets the variable operational_level=SFM. Additionally, the parameter $t_{delay}$ is introduced to provide better control over the transition by causing pedestrians to stand still for a short period after exiting the high-density zone.

**Exit choice behaviour**: Once pedestrians meet the transition criteria, they probabilistically select one of the two available exit roads. The probability of choosing an exit road depends on the distance to each of the exit roads at that time:If pedestrians are closer to Constantine Road, they choose Constantine Road as the exit road with the probability of α and Chenavard Road with the probability of 1−α.Conversely, if pedestrians are closer to Chenavard Road, they choose Chenavard Road as the exit road with the probability of β and Constantine Road with the probability of 1−β.

It is important to acknowledge that combining microscopic models and macroscopic models often requires further aggregation or disaggregation of individuals to convert and transform individual information between these models. However, our hybrid model uses a mesoscopic model and a microscopic model, both of which retain representation at the individual level. Therefore, our model sidesteps this additional step as individual information such as position and velocity are preserved when they transition to a new zone.

### 4.4. Model Calibration

Calibration of the framework is needed to realistically reproduce the pedestrian outflow of the crowd exiting during the Festival of Lights. Three types of parameters in the framework are specified for calibration:Hybridisation: $t_{delay},\alpha ,\beta$.Parameters of SFM: $A,B,V^{pref},\tau$.Parameters of CC: $f_{min},f_{max},\rho_{min},\rho_{max}$.

The Genetic Algorithm (GA) is used to calibrate the framework due to the large and non-linear search space. The objective fitness function is calculated as mean normalised absolute error:(8)$$\begin{array}{c} fitness=\frac{1}{N}\sum_{i=1}^{N}{\left(\frac{|f_{i}^{obs}-f_{i}^{sim}|}{f_{i}^{obs}}\right)}_{Constantine}+{\left(\frac{|f_{i}^{obs}-f_{i}^{sim}|}{f_{i}^{obs}}\right)}_{Chenavard} \end{array}$$
where $f^{obs},\;f^{sim}$ are the observed and simulated pedestrian outflow, respectively. The objective fitness function measures the similarity between the simulated and observed outflow across *N* observations, with lower fitness values indicating better agreement. The GA is operated through the following steps:**Initialisation**: A population of 128 individuals is initialised, with each individual representing a chromosome consisting of 11 genes corresponding to 11 parameters for calibration. Each parameter gene was initialised via random sampling from a defined range of minimum to maximum values specific to that parameter. The value interval of the parameters for each pedestrian-simulation model is chosen based on settings commonly used in the literature [7,15]. Let $\phi_{i}$ denote the *i*th parameter gene in each chromosome, where $\phi_{i}^{min}$ and $\phi_{i}^{max}$ are the minimum and maximum allowable values of the parameter $\phi_{i}$, respectively. Table 1 depicts the range of values for the parameters.**Fitness evaluation**: The fitness of individuals in the population is evaluated by comparing the simulated outflow data, extracted from simulations of a crowd exiting at the Festival of Lights, with observed outflow using the fitness function in Equation (Equation 8). A total of N=20 observations are selected via systematic sampling. Implementation details of the simulations are described in the next section.**Selection**: This process refers to choosing individuals with the best fitness values from the population to serve as parents for generating offspring for the next generation. In this work, 50% of individuals with the lowest fitness values in each generation are selected as parents for crossover and retained for the next generation.**Crossover**: Pairs of parent individuals in the best-selected set are combined to reproduce offspring. Uniform crossover is used with each gene of an offspring’s chromosome having a 0.5 probability of originating from the corresponding gene of either parent, as shown in Figure 8.**Mutation**: The mutation operator randomly alters chromosomes to prevent converging to a locally optimal solution. A mutation rate of 0.01 is applied to each gene in a chromosome. If a mutation occurs for the parameter gene $\phi_{i}$:
(9)$$\phi_{i}=\phi_{i}^{min}+\gamma \left(\phi_{i}^{max}-\phi_{i}^{min}\right),\;0\leq \gamma \leq 1.$$
where γ is a random value between 0 and 1.

Figure 9 presents the flow of the calibration process executed using the GA. An individual, which represents a set of parameters, is input into the crowd exiting simulation on the GAMA platform [30] to evaluate fitness. The GA stops when the number of generations exceeds 150.

### 4.5. Simulation Details

The experiment configuration for simulating the crowd exit scenario is as follows:The simulation is conducted with a time step Δt=0.1 s, with 3883 agents.Pedestrians are initialised with a uniform distribution across the high-density zone.All simulations are conducted using the GAMA platform [30] on a M1 MacBook Pro with 32 GB of memory.

### 4.6. Calibration Results

Figure 10 shows the average fitness value and best fitness value of the population over 150 generations. It can clearly be seen that the average fitness exhibits an exponential decrease in the first 10 generations. Then, it declines gradually until generation 35, where it stabilises for the remaining duration. Similarly, the best fitness value of the population drops rapidly until generation 30 before reaching a stable state.

Table 2 presents the optimal parameter values obtained through the calibration process. The calibrated α and β values indicate that pedestrians are more likely to choose the closest road to exit. However, β is lower than α. This can be explained by the fact that there are more pedestrians near Chenavard Road than Constantine Road at the time when the exit choice behaviour is performed.

### 4.7. Model Validation

The calibrated parameters are used to reproduce the pedestrian outflow of the crowd exiting at the Festival of Lights. A total of 90 simulations are conducted with the calibrated parameters. The simulated outflow is extracted and the average is computed with a 95% confidence interval. Figure 11 presents a comparison of simulated outflow and observed outflow for the two exit roads. As illustrated, the simulated outflow exhibits close alignment with the observed outflow data.

### 4.8. Sensitivity Analysis

It is important to understand how different parameters in the calibrated model can influence the simulation results. A local sensitivity analysis is therefore conducted by individually varying the value of one parameter at a time around its calibrated value while holding all other parameters constant. The values vary from −25% to 25% in increments of 5% as long as they fall within their corresponding permissible ranges, except for $t_{delay}$ because of its dependence on discrete time step value. For each combination of parameter values, 40 simulations are run to compute the average pedestrian outflow and its 95% confidence interval.

Figure 12 presents the local sensitivity analysis results for each parameter. For all hybridisation parameters, the fitness function generally reaches the optimal value at the calibrated value of each parameter. The parameter β demonstrates the strongest sensitivity to the simulation results, with the fitness value increasing sharply as β moves away from 0.75. In contrast, although $t_{delay}$ and α display similar trends, their effects on fitness values are less pronounced than those of β.

On the other hand, for the parameters of the pedestrian-simulation models, the sensitivity results differ between the SFM and CC models. For the parameters of the SFM, the fitness values remain largely unchanged, indicating that the SFM parameter variations do not substantially affect the fitness value and the outflow results. For the parameters of the CC model, only those parameters specifying pedestrian speed, including $f_{min}$ and $f_{max}$, exhibit strong sensitivity to the fitness values, with a rapid increase when these parameters deviate from their calibrated values. In contrast, the parameters $\rho_{min}$ and $\rho_{max}$ have almost no impact on the fitness value.

## 5. Performance Analysis

This section demonstrates the performance of the hybrid model in simulating a large number of pedestrian agents. As such, the hybrid model is compared to three others:SFM-only model: the SFM is applied in all three zones.3-CC-1 model: consists of three separate CC models, each with one target cell for simulating one single zone.1-CC-2 model: one CC model for simulating the entire environment, with two designated target cells representing two exit roads. This configuration increases computational complexity compared to using only one target cell.

For each simulation, two indicators are computed, and an average is determined from a total of 15 simulations for each model:Density map (in ped/m^2^) of pedestrian density distribution across the simulation area.Computation time (in s) required to calculate one simulation step.

Figure 13 presents the density maps of various models over different time steps (*t* = 60 s, 150 s, 240 s). In the SFM-only model, the highest density area, with a maximum density of 3.5 ped/m^2^, is observed in both the plaza and the two exits. Similarly, the 3-CC-1 model and 1-CC-2 model can simulate extremely high densities of 6–8 ped/m^2^, but these extremely high-density areas also appear in both the plaza and the two exits. In contrast, the hybrid model exhibits a clear difference in density levels between the plaza and the two exits. These results indicate that the hybrid model can effectively simulate pedestrians in environments with a mix of low- and high-density situations. Furthermore, a key advantage of this model is its generic nature and flexibility, as it can accommodate any combination of zones and models, enabling the modelling of various scenarios and crowd dynamics.

Next, various numbers of pedestrian agents (3 K to 15 K) are used to evaluate the simulation time of different models. Figure 14a displays the simulation of 6000 agents at Place des Terreaux using the hybrid model, with pedestrians in the high-density zone represented in red and pedestrians in the low-density zones shown in blue. The results, shown in Figure 14b, indicate that the SFM-only and 3-CC-1 models have the longest simulation times, which increase significantly as the number of pedestrians grows, while the 1-CC-2 model has a similar simulation time to them at first but drops as the number of agents increases. In contrast, our model demonstrates superior performance in terms of simulation time, with a large difference from the other models observed as the number of pedestrians grows. This finding indicates that using mesoscopic models for low-density regions is unnecessary, and they can be replaced with microscopic models. These results suggest that having different modelling algorithms at each level not only improves the variety of behaviour observed in various scenarios but also enhances overall performance compared to using only microscopic models or multiple mesoscopic models.

## 6. Conclusions and Discussion

Large-scale crowd phenomena are complex to model as the behaviour of pedestrians needs to be described at strategic, tactical, and operational levels and it is impacted by the density of the crowd. Particularly in the operational level modelling, each model can only simulate a single type of pedestrian dynamic, while pedestrian dynamics vary significantly depending on the density of the environment. This paper proposes an agent-based model for simulating the multi-level behaviour of pedestrians, where agents are able to dynamically change their operational model based on local density. The ability to combine different models of the proposed agent-based model for multi-scale simulation is evaluated through a use case of pedestrians at the Festival of Lights, Lyon, France.

The hybrid model is calibrated using a genetic algorithm that utilises pedestrian outflow data extracted from video recordings of exiting crowds at the festival. Simulation results of the calibrated model show a match between simulated and observed pedestrian outflow over time. Furthermore, a local sensitivity analysis is then conducted to identify the most sensitive parameters in the model. Finally, the performance of the hybrid model is compared to different models in terms of density map and computation time. The comparison results demonstrate that the hybrid model has the capacity to effectively simulate pedestrians across varied density scenarios while optimising computational performance compared to other models.

The proposed framework is useful for simulating a broad range of pedestrian dynamics through its capacity to integrate various pedestrian-simulation models and their coupling. The combination of these models can be used for large-scale and multilayered simulations of crowds across numerous scenarios. Additionally, the framework facilitates comparative analysis of the performance of different pedestrian-simulation models. Moreover, the proposed framework is not only generic in the domain of pedestrian modelling but also extensible to other related domains such as traffic simulation and social simulation.

Our future work aims to incorporate a greater variety of modelling algorithms for each level of behaviour, particularly at the operational level. Then, the selection of an optimal algorithm at each level of behaviour and the criteria for switching these algorithms must be defined based on the density of each region. Another promising direction involves applying density-based clustering algorithms to pedestrian coordinates in order to dynamically estimate the local density of environments.

## Figures and Tables

**Figure 1 sensors-24-01639-f001:**
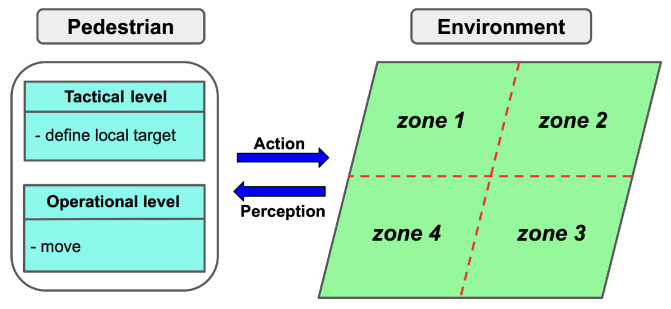
General overview.

**Figure 2 sensors-24-01639-f002:**
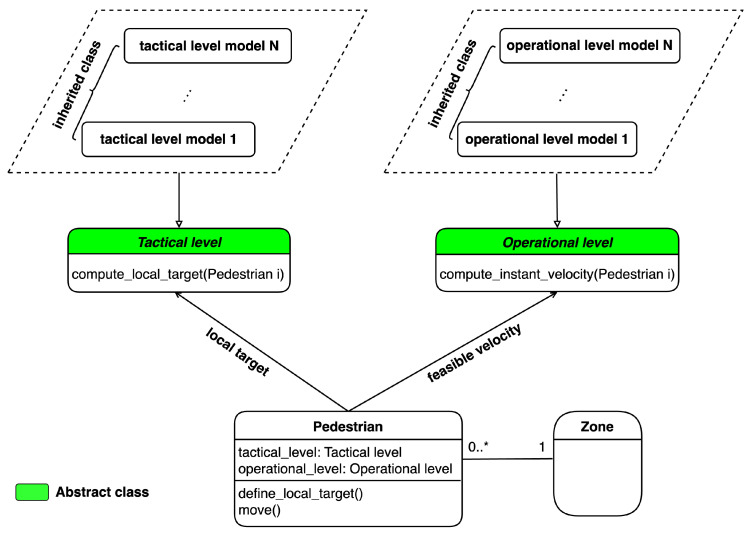
Agent-based model for multi-level pedestrian behaviour.

**Figure 3 sensors-24-01639-f003:**
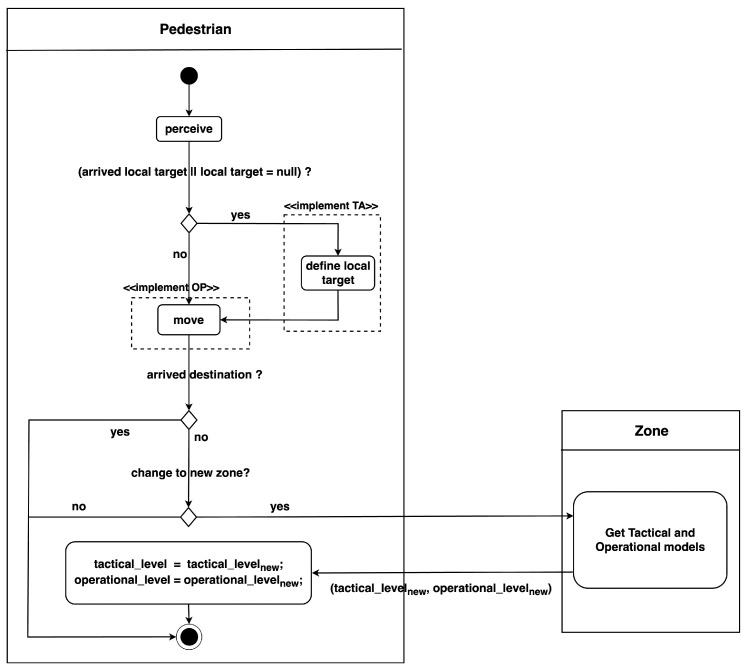
Pedestrian activity diagram at each simulation step.

**Figure 4 sensors-24-01639-f004:**
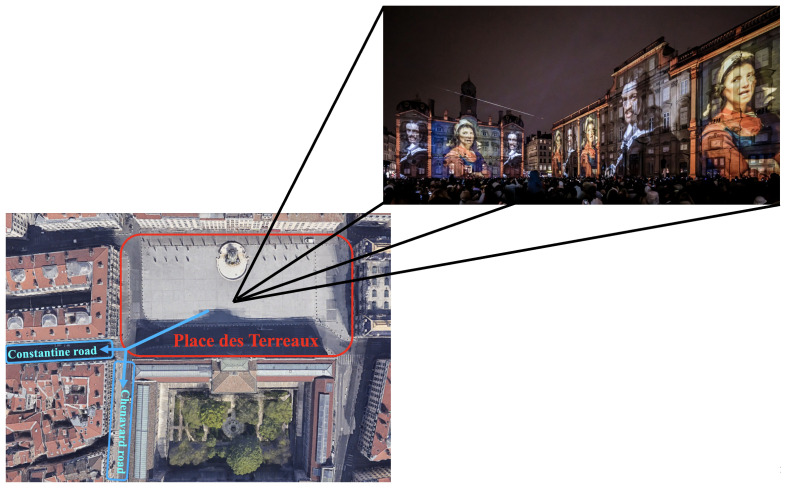
A screenshot of pedestrians watching the show at the Place des Terreaux in 2022 (https://www.fetedeslumieres.lyon.fr/fr/oeuvre/grand-mix-au-musee-des-beaux-arts-de-lyon (accessed on 30 January 2024)) and main circulation of the exiting crowd.

**Figure 5 sensors-24-01639-f005:**
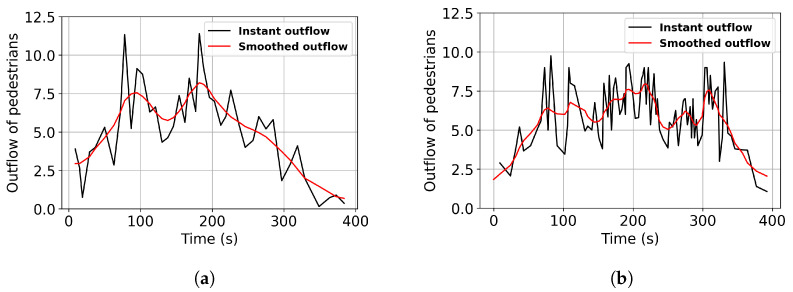
Empirical outflow of pedestrians. (**a**) The outflow of Constantine Road. (**b**) The outflow of Chenavard Road.

**Figure 6 sensors-24-01639-f006:**
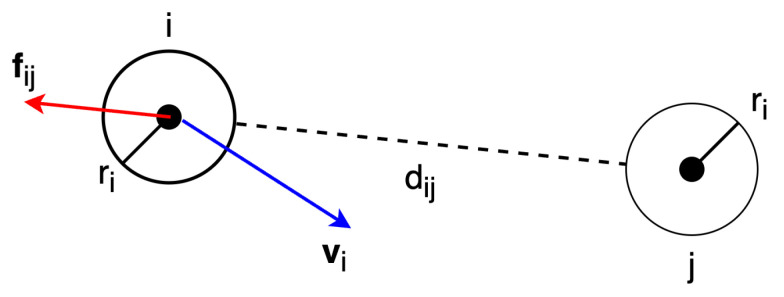
Description of repulsion force $f_{ij}$ between pedestrian *i* and *j*.

**Figure 7 sensors-24-01639-f007:**
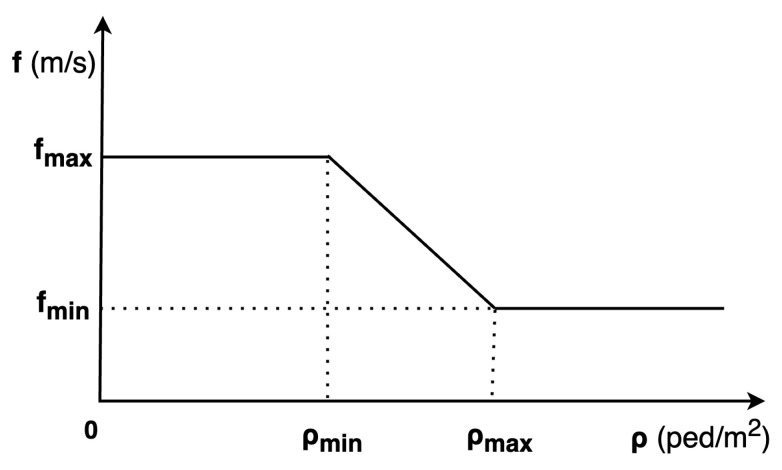
Parameters of CC model.

**Figure 8 sensors-24-01639-f008:**
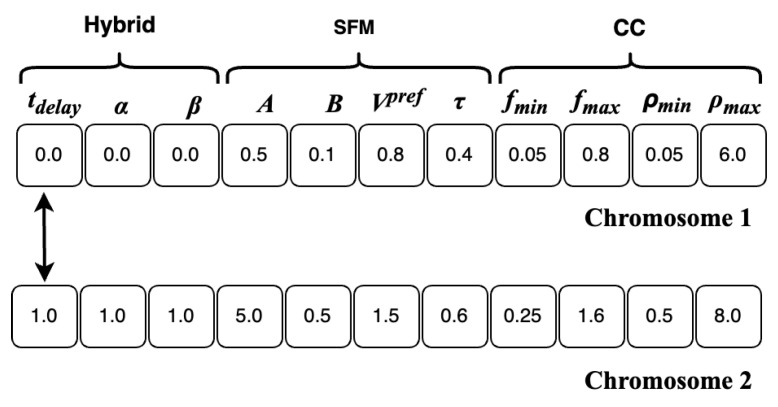
Crossover.

**Figure 9 sensors-24-01639-f009:**
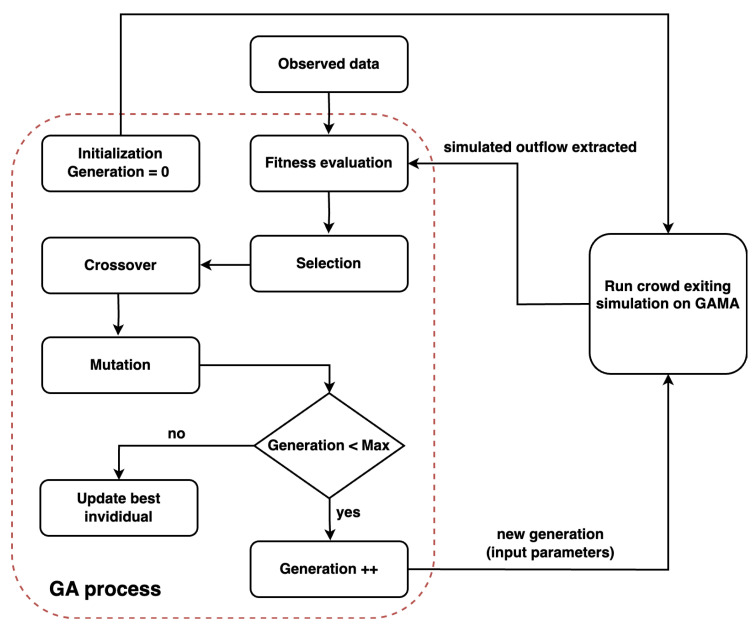
Flow of the calibration process.

**Figure 10 sensors-24-01639-f010:**
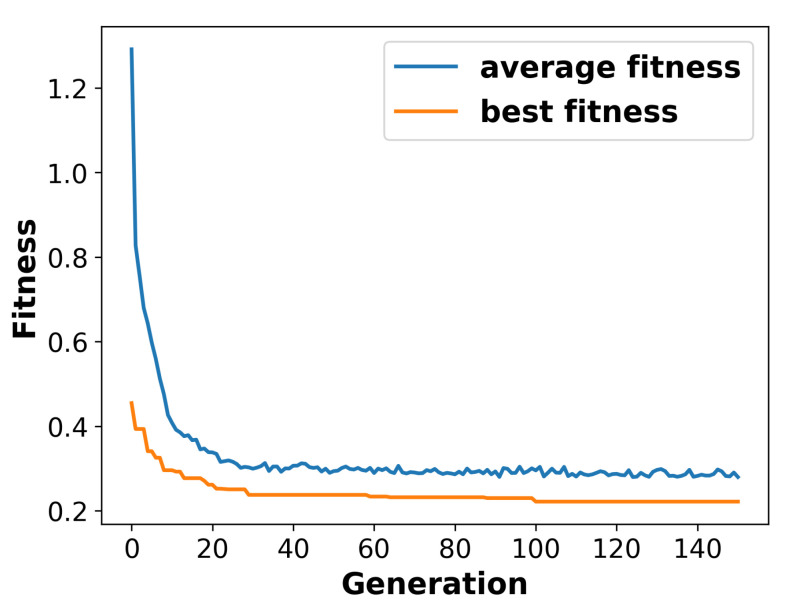
Average and best fitness values over the generations.

**Figure 11 sensors-24-01639-f011:**
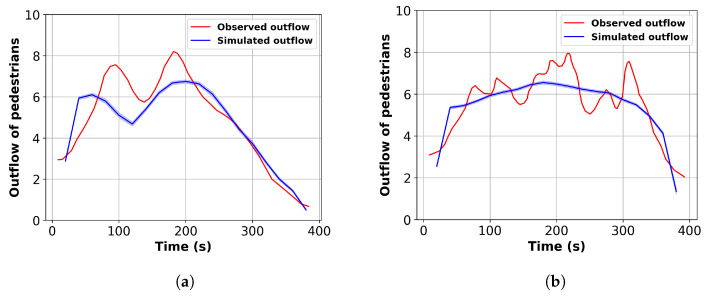
Comparison of observed and simulated outflow for the two exit roads. (**a**) Constantine Road. (**b**) Chenavard Road.

**Figure 12 sensors-24-01639-f012:**
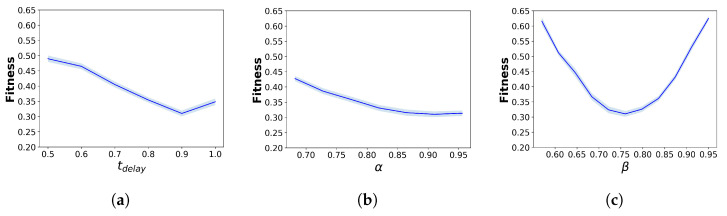
Local sensitivity analysis for different parameters. (**a**) $t_{delay}$, (**b**) α, (**c**) β, (**d**) *A*, (**e**) *B*, (**f**) $V^{pref}$, (**g**) τ, (**h**) $f_{min}$, (**i**) $f_{max}$, (**j**) $\rho_{min}$, (**k**) $\rho_{max}$.

**Figure 13 sensors-24-01639-f013:**
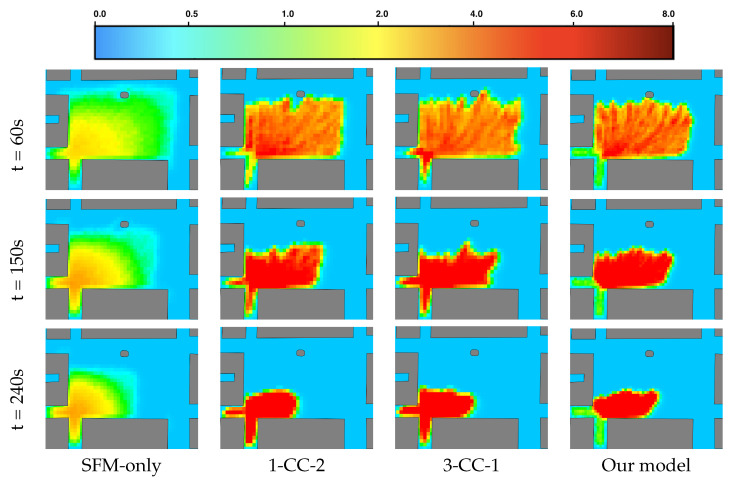
Density maps among different models with 6000 simulated agents.

**Figure 14 sensors-24-01639-f014:**
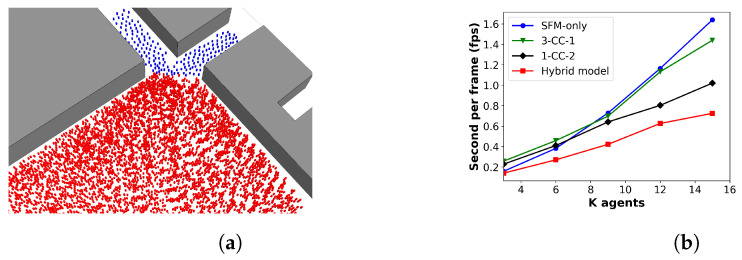
Simulation results. (**a**) Simulation of 6000 agents at the Place des Terreaux using the hybrid model with pedestrians in the high-density zone in red and those in the low-density zones in blue. (**b**) Comparison of performance for different models.

**Table 1 sensors-24-01639-t001:** List of parameters and their ranges.

Type	Parameter $\phi_{i}$	$\phi_{i}^{\min}$	$\phi_{i}^{\max}$
**Hybrid**	$t_{delay}$	0.0	1.0
	α	0.0	1.0
	β	0.0	1.0
**SFM**	*A*	0.5	5.0
	*B*	0.1	0.5
	$V^{pref}$	0.8	1.5
	τ	0.4	0.6
**CC**	$f_{min}$	0.05	0.25
	$f_{max}$	0.8	1.6
	$\rho_{min}$	0.05	0.5
	$\rho_{max}$	6.0	8.0

**Table 2 sensors-24-01639-t002:** Optimal parameter values obtained via the calibration process.

**Parameter** $\phi_{i}$	$t_{delay}$	α	β	*A*	*B*	$V^{pref}$	τ	$f_{min}$	$f_{max}$	$\rho_{min}$	$\rho_{max}$
**Best value**	0.9	0.91	0.76	1.83	0.45	1.25	0.57	0.15	1.35	0.11	6.36

## Data Availability

The data that support the findings of this study are available from the corresponding author, Huu-Tu Dang, upon reasonable request.

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
