# Peer review of "HyPedSim: A Multi-Level Crowd-Simulation Framework—Methodology, Calibration, and Validationâ€"

_sensors, 2024, doi:10.3390/s24051639_

Round 1
Reviewer 1 Report
Comments and Suggestions for Authors
The paper provides interesting and valuable results.
There are some minor issues that should be corrected
line 58 typo: "related" (not "relate")
line 104 - Fig 1 looks like some of the environment is not a zone. However, the text reads "The environment is divided into different zones," which means that any any piece of the environment belongs to this or that zone. Please clarify.
lines 104-108 examples of zones are needed
lines 105-106 "An appropriate model is selected to simulate pedestrians in each zone depending on crowd density" - it's not clear, what are these models are. Please consider placing desription of the models here
Formulas (1),(2) look like all the variables are scalars. I guess, not.
line 126 I believe, every time the word "class" appears in new context, it should be specified: class of what?
line 138
what are "the variables t and o in Pedestrian class"? I can only see the uppercase "O" capital, not lowercase "o". Then, lowercase "t" is not actually what is meant here, but the time in Eq. (1), (2).
Fig. 2 What is nav_mesh? I also recommend explaining right here what are CC and SFM. It's not good when abbrevationes are explained three pages after they are used the first time.
Reviewer 2 Report
Comments and Suggestions for Authors
Dear colleagues,
Your article is interesting. But in order to make it more interesting for real system developers try to correct it taking into account the following remarks.
Remarks:
1. It remains unclear who are model stakeholders and what concerns they have.
2. For modeling multi-level behavior, the authors suggest using an agent-based approach, while the use of this approach is not argued in any way. The question arises about the possibility of using other approaches, e.g. digital twins or high level architecture.
3. The first keyword is the term 'Agent-based architecture', but there is no architectural description in the article (see ISO/IEC/IEEE 42010:2011).
4. It remains unclear whether the authors suggest using the proposed models in runtime mode or in post mortem mode.
5. It would be necessary to define the scope of the proposed models usage somehow.
6. The authors propose to use a hierarchy of models, but do not define the ways of representing models at separate levels and the mechanisms of their transformation (fusion).
Best regards,
